# Cytosolic Self-DNA—A Potential Source of Chronic Inflammation in Aging

**DOI:** 10.3390/cells10123544

**Published:** 2021-12-15

**Authors:** Mansour Akbari, Daryl P. Shanley, Vilhelm A. Bohr, Lene Juel Rasmussen

**Affiliations:** 1Department of Cellular and Molecular Medicine, Center for Healthy Aging, SUND, University of Copenhagen, 2200 Copenhagen, Denmark; BohrV@grc.nia.nih.gov; 2Campus for Ageing and Vitality, Biosciences Institute, Newcastle University, Newcastle Upon Tyne NE4 5PL, UK; daryl.shanley@newcastle.ac.uk; 3Section on DNA Repair, National Institute on Aging, 251 Bayview Blvd, Baltimore, MD 21224, USA

**Keywords:** aging, inflammation, cGAS-STING, DNA repair, cytosolic self-DNA, mitochondria

## Abstract

Aging is the consequence of a lifelong accumulation of stochastic damage to tissues and cellular components. Advancing age closely associates with elevated markers of innate immunity and low-grade chronic inflammation, probably reflecting steady increasing incidents of cellular and tissue damage over the life course. The DNA sensing cGAS-STING signaling pathway is activated by misplaced cytosolic self-DNA, which then initiates the innate immune responses. Here, we hypothesize that the stochastic release of various forms of DNA from the nucleus and mitochondria, e.g., because of DNA damage, altered nucleus integrity, and mitochondrial damage, can result in chronic activation of inflammatory responses that characterize the aging process. This cytosolic self-DNA-innate immunity axis may perturb tissue homeostasis and function that characterizes human aging and age-associated pathology. Proper techniques and experimental models are available to investigate this axis to develop therapeutic interventions.

## 1. Introduction

Studies in human materials and in animal models have consistently demonstrated that a state of systemic, low-grade, chronic inflammation, develops in various tissues with advancing age [1,2,3,4,5,6,7]. Aging is the biggest risk factor for increasing incidence of chronic diseases such as type 2 diabetes, cardiovascular diseases, kidney disease, cancer, and neurodegenerative diseases. Chronic inflammation is a common condition in all those diseases [1].

Frailty commonly refers to a clinical state associated with increased vulnerability in older adults to risk of injury, disability, hospitalization, and mortality [8]. Frailty is closely associated with inflammation and high levels of inflammatory markers including C-reactive protein and interleukin-6 have been reported in the pro-frail and frail elderly [9].

Processes that initiate and prolong age-associated inflammation likely involve various tissue and cell type dependent mechanisms that are further influenced by inherited genetic predisposition and lifestyle [1,10]. The age-associated chronic inflammation damages tissues and organs and increases susceptibility to disease [11,12].

The aging process is defined as an overall progressive decline in physiological function over time associated with an increasing risk of disease and death. At the molecular level, aging is the consequence of a lifelong accumulation of stochastic damage to tissues and cellular components [13,14].

Homeostasis is the active maintenance of defined quantitative variables characteristics of the cells, tissues and an organism within a desired range [15]. In this context, age-related altered chronic inflammatory responses can be explained as a result of disturbances in cellular and tissue homeostasis, in part, because of persistent cellular and tissue damage [15].

Inflammation can also be initiated by conditions that are often prelude to homeostasis disturbances like misplaced cellular components [15]. Misplaced cytosolic nuclear and mitochondrial DNA, e.g., as a result of impairment in the systems that maintain genome stability or those that are involved in mitochondrial quality control, have emerged as potent inflammation promoting cellular components by activating the innate immune system mediated by specific nucleic acid sensors as damage-associated molecular patterns [16].

The importance of proper cellular response to cytosolic DNA is evidenced in patients with autoimmune disease caused by mutations in enzymes that prevent the accumulation of cytosolic DNA [17], or pathogenic mutations that lead to constitutive activation of the signaling systems that respond to cytosolic DNA [18]. These subjects are more closely discussed later.

Here, we discuss the documented and inferred role of the cytosolic self-DNA as a potential source of age-related inflammation. This Review is not an extensive description of these pathways. Sources of in-depth information are included for further reading.

## 2. Cytosolic DNA Sensors

From single cell organisms to mammals, different types of sensors have evolved to detect nucleic acids of invading pathogens and subsequently activate a defense response in the infected cell [16,19,20,21,22,23,24,25,26,27].

Several cytosolic DNA sensors have been identified including absent in melanoma 2 (AIM2), cyclic GMP-AMP synthase (cGAS), interferon-γ-inducible 16 (IFI16). Following binding to cytosolic DNA, these sensors activate various inflammatory responses [28].

AIM2 activation is best known to result in the assembly of the inflammasome (e.g., NLRP3), followed by activation of inflammatory caspases that trigger pyroptosis, a proinflammatory form of cell death [29].

IFI16 senses cytosolic DNA from different sources and is able to activate IFNI as well as to trigger inflammasome assembly and pyroptosis [30,31,32]. In human keratinocytes, IFI16 was found to functionally interact with cGAS for full activation of innate immune response to exogenous DNA and DNA viruses [33]. Moreover, activated IFI16 promoted STING phosphorylation and transport [33]. These results demonstrate that both the IFIT16 and cGAS DNA sensors trigger IFNI response through STING activation.

Here, we focus on cGAS-STING signaling because of its emerging central function in cellular inflammatory response to misplaced cytosolic self-DNA in both immune and non-immune cells.

### The cGAS-STING Pathway

The cytosolic DNA sensor cyclic GMP-AMP Synthase (cGAS) and its effector protein Stimulator of Interferon Genes (STING) initiate an important signaling pathway in the defense against microbial infection and inflammation [21,23,34,35].

cGAS is activated following binding to double-stranded DNA originating from the infecting microbes as well as from misplaced self-DNA from the nucleus and mitochondria in both immune and non-immune human cells. Activated cGAS produces the second messenger cGAMP [23], which binds to STING resulting in the transport of STING from the endoplasmic reticulum to the trans-Golgi network where it activates TKB1 kinase and transcription factor IRF3. These interactions promote the expression of type I interferons (IFNI), and inflammatory cytokines and chemokines [36,37] (Figure 1). Moreover, gap junctions and cGAMP transporters allow rapid flow of cGAMP between adjacent cells [38,39,40]. Thus, accumulation of cytosolic DNA in one cell can activate inflammatory responses in multiple adjacent cells thus amplifying a local tissue inflammatory response (Figure 1). cGAS-STING pathway activation and STING transport from the endoplasmic reticulum to the Golgi are largely controlled by post-translational protein modifications, protein-protein interactions, and higher order spatiotemporal organization [18,41,42,43,44,45,46]. Moreover, cGAS-STING activation and response seems to be controlled in a feedback loop manner. Both cGAS and STING are reported as interferon stimulated genes (ISG) and their expression have been shown to be positively controlled by IFNI [47,48], thus enhancing the cGAS-STING response to cytosolic DNA. Mammalian cells have several systems in place to prevent the accumulation of DNA molecules in cytoplasm and the subsequent uncontrolled and persistent harmful inflammation [19]. The cytosolic Three prime repair exonuclease 1 (TREX1) is a key enzyme for clearing cytosolic DNA [49]. Mutation in TREX1 causes Aicardi-Gautier Syndrome, a disease characterized by constitutive IFNI production and severe neuropathy [50]. In Trex^−/−^ mice, deletion of cGAS, STING, or IRF3, abrogates this chronic inflammatory state [17,51,52]. Thus, various forms of self-DNA are frequently released into cytoplasm that can activate the cGAS-STING signaling and IFNI production unless they are rigorously degraded [53].

## 3. Sources of Cytosolic Self-DNA

### 3.1. Nuclear DNA

The preservation of the chemical structure and the nucleotide sequence of the genome are essential for life. Thus, all living organisms have evolved mechanisms to repair DNA lesions and to maintain genome stability and integrity [54]. Evidence indicates that DNA damage and mutation accumulate with age in multiple tissues across species [55,56,57,58,59]. DNA damage and genome instability can contribute to aging in various ways, for instance, by perturbing normal gene expression, persistent and elevated DNA damage response signaling, apoptosis, and cellular senescence [56].

Genome instability and elevated markers of inflammatory responses are consistent features of aging tissues [1,2,60]. A plausible connection between genome instability and inflammation is the growing evidence that DNA damage and replication stress can promote cytosolic mislocalization of nuclear DNA and the subsequent activation of the cGAS-STING pathway [36,61].

#### 3.1.1. Micronuclei

Micronuclei are extra-nuclear small nuclear membrane enclosed structures with whole or fragments of chromosomes [62]. An accurate and highly coordinated process of chromosome segregation during cell division is necessary to ensure that each daughter cell receives a complete and identical copy of the parental genome. Micronuclei formation is tightly connected to defective mitosis and indicates chromosomal instability [63].

Ultrafine anaphase bridges are DNA molecules that connect the separating sister chromatids in the anaphase of mitosis. PLK1-interacting checkpoint helicase (PICH) is a DNA translocase necessary for the resolution of ultrafine anaphase DNA bridges to ensure the fidelity of chromosomal segregation. Cells lacking PICH display chromosomal instability and increased level of micronuclei formation [64].

A considerable number of ribonucleotides are inserted into the genome during replication, which if not properly repaired can give rise to genome instability [65]. RNAse H2 is an important enzyme for removal of ribonucleotides from DNA [65]. Loss of function mutations in RNase H2 cause Aicardi-Gautier Syndrome and interferonopathy, defined as a constitutive upregulation of IFNI [50,66], which is cGAS-STING dependent [67]. Evidence suggests that RNase H2 deficiency promotes micronuclei formation followed by enhanced expression of interferon stimulated genes, ISGs, that involves the cGAS-STING signaling pathway [68].

RNA:DNA hybrids can occur during DNA transcription and replication and are formed both in the nucleus and in mitochondria [69]. RNA:DNA hybrids appear to accumulate in RNase H2-deficient Aicardi-Gautier Syndrome patient cells and elicit INFI expression [66]. Fibroblasts from centenarians were found to contain high levels of RNase H2, concomitant with low levels of cytoplasmic RNA:DNA hybrids, a RNase H2 substrate, and markers of pro-inflammatory responses [70]. This may reflect that centenarians may have increased repair of highly frequent ribonucleotide DNA lesions.

BLM belongs to the family of RecQ helicases and is essential for the maintenance of genome stability [71]. Defects in BLM in humans gives rise to Bloom syndrome characterized by cancer predisposition and premature aging [71]. A higher than normal incidence of micronuclei is characteristic of BLM deficient cells [72].

Micronuclei are not stable structures because they cannot import all the proteins that are necessary for the preservation of the nuclear envelope integrity [73]. Occasional disintegration of the micronuclei membrane exposes their DNA content to cGAS and the subsequent induction of the IFNI response [74].

Micronuclei formation reportedly increases with age and appear to occur often in several accelerated aging syndromes [75,76,77], and in cancer [78]. Age-associated decline in mitotic fidelity and function may underlie increased micronuclei formation over time [76]. Thus, micronuclei may present a mechanistic link between genome instability and innate immune activation, two hallmarks of aging tissues.

#### 3.1.2. Retrotransposable Elements

Transposons are genetic elements that can translocate from one genome locus to another. Long interspersed nuclear elements (LINEs; L1) are actively replicating retrotransposable elements that use reverse transcription of RNA intermediates to produce double-stranded DNA products to amplify and be inserted into new positions in the genome [79].

Retrotransposon DNA sequences are abundant in the human genome [80]. Transcriptional activation of retrotransposons and their subsequent reverse-transcription to cDNA molecules in the cytoplasm, constitute a potential source of nucleic acid evoked immune responses and IFNI expression [81,82].

Activation of retrotransposons in mammalian cells is thought to be a rare event likely due to various control mechanisms including epigenetic transcription suppression [83]. Aging is associated with global alteration in the level of epigenetic DNA methylation and chromatin remodeling that can contribute to de-repression of transcription of retroelements and the subsequent chronic immune activation in aging tissues [84,85,86].

### 3.2. Mitochondrial DNA

Mitochondria are the primary sites for cellular ATP production and play a central role in key cellular processes, including apoptosis, cytoplasmic calcium buffering, and reactive oxygen species (ROS) mediated signaling pathways [56]. Because of the central position of mitochondria in these key cellular processes, impaired mitochondrial homeostasis is commonly recognized in connection to multiple aspects of the aging process and the age-related onset of frailty and diseases [56].

Each mitochondrion contains multiple copies of the 16.6 kb circular mitochondrial DNA (mtDNA) packaged with mitochondrial transcription factor A (TFAM) and other proteins into structures called nucleoids [87]. Accumulation of various forms of mtDNA damage correlates with aging and age-related diseases [60]. This indicates that diminished mtDNA integrity is tightly connected to tissue aging and that mtDNA is highly susceptible to damage accumulation in the course of life.

It has long been known that mtDNA released from mitochondria can elicit inflammatory responses [88,89]. In the following sections, we discuss documented and inferred mechanisms for cytosolic misplacement of mtDNA.

#### 3.2.1. Impaired Mitochondrial Dynamics

Mitochondria form interconnected networks and undergo fusion and fission that are essential for proper mitochondrial and cellular function [90,91]. Fusion and fission, and mitochondrial quality control by autophagy, are interconnected processes that collectively define mitochondrial morphology and dynamics [91,92]. Reports suggest that these processes are influenced by age across species [93,94,95].

Dynamin-related protein 1 (DRP1) is a key regulator of mitochondrial fission. Increased mitochondrial fragmentation due to DRP1 over-activation promotes cytosolic mtDNA levels, as determined by PCR analysis of mtDNA, followed by cGAS-dependent inflammatory response [96]. Optic atrophy protein 1 (OPA1) is an inner mitochondrial membrane protein that regulates mitochondrial fusion and cristae structure remodeling, and is involved in the preservation of mtDNA integrity [97,98,99,100]. Muscle-specific depletion of OPA1 deletion in mice causes muscle inflammation that is dependent on mtDNA probably because an altered mitochondrial structure promotes mtDNA release [101].

YME1L is an ATP-dependent protease embedded in the mitochondrial inner membrane. It controls fusion and fission of mitochondria by processing OPA1, and controls de novo nucleotide synthesis and pyrimidine uptake into mitochondria [102]. Homozygous mutation in the YME1L1 gene causes infantile-onset mitochondriopathy associated with intellectual disability, motor developmental delay, optic nerve atrophy associated with visual impairment, and hearing impairment [103].

In mice, loss of YME1L in the nervous system causes inflammatory responses in the retina [104], which may, in part, be caused by an increased release of mtDNA into cytoplasm and a cGAS-STING dependent inflammatory response [105].

Collectively, these reports show that disturbances in the processes that control mitochondrial structure and quality may contribute to the release of mtDNA into the cytoplasm.

#### 3.2.2. Mitochondrial Stress

Mitochondria are a key source of cellular ROS production. These molecules are important in redox signaling but can also contribute to mitochondrial damage and stress. Mitochondrial abnormality is often associated with elevated levels of ROS and free radicals that can generate oxidative damage to tissues and cellular components [106]. Mitochondria contain a number of enzymatic and non-enzymatic antioxidant systems to control and neutralize ROS and free radicals, which reportedly diminishes with advancing age [106]. Mitochondrial ROS production has long been considered a significant driving force in the aging process, a concept known as the “free radical theory” of aging [107]. Emerging evidence, however, depicts a nuanced role of ROS on life span [108,109].

Various conditions that cause cellular and mitochondrial stress may also promote the release of mtDNA and the activation of immune responses. Increased mitochondrial stress in TFAM-deficient mice led to mtDNA release followed by activation of cGAS and IFNI expression [110]. Permeabilization of the mitochondrial membrane is known to facilitate mtDNA release [111]. Melatonin is a mitochondrial antioxidant. Melatonin levels reportedly decline with age [112]. Experimental ablation of melatonin in mouse resulted in the release of mtDNA and cGAS-STING activation due to reduced mitochondrial membrane potential and increased mitochondrial permeability [113]. The oligomerization of the mitochondrial outer membrane protein, voltage-dependent anion channel, in cells undergoing oxidative stress promotes pore formation and mtDNA release [114].

Exosomes are membrane-bound extracellular vesicles that contain protein, DNA and RNA molecules of the cells that secrete them. In cells undergoing prolonged, low-level oxidative stress, mtDNAs were transported out of the cell in exosomes that were subsequently transported into the adjacent cells where they induced inflammation [115]. Thus, mtDNA release in a cell undergoing stress can propagate to bystander cells and amplify inflammatory responses locally and in distant cells through exosomes secretion and uptake.

Accumulation of various forms of protein aggregates in neurons is characteristic of a number of neurodegenerative diseases but it is also somewhat prevalent in physiological aging brains [5]. Protein aggregates can interrupt mitochondrial function and generate mitochondrial stress, for instance, by entering mitochondria and interfering with mitochondrial biogenesis [116,117,118,119].

Aging is the primary risk factor for most neurodegenerative diseases including Alzheimer disease (AD). Conditions that intimately associate with AD brains include amyloid β plaques, tau oligomerization and aggregation, mitochondrial dysfunction, and inflammation [120]. In AD brains, intraneuronal accumulation of soluble tau oligomers correlates with synaptic alterations, neuroinflammation and memory deficits [121]. Hippocampal neurons in a mouse model of human tauopathy, displayed cellular stress and altered mitochondrial morphology and homeostasis [119], as well as, T cell infiltration and inflammation that associated with cognitive decline [122].

Neuronal cytoplasmic inclusions of the nuclear TAR DNA-binding protein 43 (TDP-43) is a characteristic of the neurodegenerative diseases amyotrophic lateral sclerosis and frontotemporal dementia. Mitochondrial translocation of TDP-43 can promote the extrusion of mtDNA via the mitochondrial permeability pore and subsequently the cGAS activation, and IFNI production [123]. Thus, although amyotrophic lateral sclerosis and frontotemporal dementia are not strictly age-related diseases, they demonstrate that cytosolic localization of mtDNA may account, at least in part, for the well-established close associations between altered mitochondrial homeostasis and biogenesis, immune signaling, and inflammation, in age-related neurodegenerative diseases [124,125,126,127,128].

Thus, disturbed mitochondrial homeostasis and cellular stress can enhance mtDNA release by various mechanisms. In addition, the following properties of mtDNA, as well as mitochondrial biology in general, support cytosolic mtDNA as a plausible trigger of chronic inflammatory response in aging tissues:(1)MtDNA is located in close association with the mitochondrial inner membrane, which is a major site of mitochondrial reactive oxygen species production. Thus, mtDNA is thought to contain high steady state levels of oxidative damage [129], which seemingly renders it resistant to degradation in cytoplasm [130]. Moreover, the circular mtDNA resembles bacterial DNA in being hypo- or unmethylated at CpG-motifs [131,132]. Together, these properties appear to render mtDNA a particularly potent immunogenic molecule [88,89].(2)Endocytosis of circulating cell-free mtDNAs elicits an inflammatory response by Toll-like receptor 9 that preferentially binds to unmethylated CpG nucleotides [88,89,133,134]. The amount of circulating mtDNA in plasma positively correlates with advancing age and may thus constitute a source of age-related systemic low-grade inflammation [135].(3)Some cell types and tissues may be more susceptible to stochastic mtDNA release. In neurons, for instance, physiological turnover of mitochondria as well as the processes of fission and fusion that maintain the mitochondrial pool largely takes place in the soma [136,137,138]. Thus, mitochondria are regularly transported to the synapsis and back through axon to soma. Perturbation in this system may result in frequent release of mtDNA. In addition, the rate of constitutive autophagosome biogenesis reportedly declines in aging neurons [139], which may diminish autophagic turnover of damaged mitochondria. Incomplete digestion of mitochondria and mtDNA during autophagy also triggers inflammatory responses [140].(4)Sheer number: Each cell contains multiple mitochondria and hundreds of mtDNA molecules. Thus, any disturbances in normal mitochondrial homeostasis such as the fission and fusion and mitochondrial turnover by autophagy, could result in frequent, stochastic release of mtDNA.

Reports on cGAS activation by cytosolic mtDNA have largely relied on cell lines and experimental settings in which the integrity of mitochondria is intentionally compromised [110,141,142]. Although these results clearly identify various mechanisms for the release of mtDNA, more research is needed to document mtDNA release under physiologically relevant conditions. Moreover, bioinformatics analysis of the genes that have been experimentally demonstrated to promote mtDNA release in various omics datasets [143] may provide some support for increased age-related cytosolic mtDNA release.

## 4. Cellular Senescence

Senescence is a state of permanent cell-cycle arrest in somatic cells that is thought to prevent proliferation of damaged cells such as those with persistent DNA damage that can develop to cancer cells [144,145,146]. Cellular senescence has long been recognized as a consistent feature of aging tissues [147]. The number of senescent cells increases in various tissues over time, which is thought to contribute to the systemic impairment in tissue function, repair, and regeneration with age. Accordingly, selective clearance of senescent cells in mice has been reported to postpone aging-related disorders [148,149,150]. However, indiscriminate elimination of senescent cells may disrupt tissue function and cause health deterioration, because it appears that defined senescent cells may have important tissue specific functions [151].

Senescence-associated secretory phenotype (SASP) is an established characteristic of senescent cells and refers to the expression and secretion of various pro-inflammatory cytokines, chemokines, proteases and growth factors that are thought to diminish tissue function and homeostasis over time [1,152,153].

The central role of DNA damage and DNA damage response in the progression of cellular senescence together with SASP has encouraged investigations into a possible role of cGAS-STING in the promotion of cellular senescence.

cGAS-STING activity appears to underlie key senescence phenotypes, in part, as a result of release of chromosome fragments into cytoplasm due to the disruption of the nuclear envelope integrity in the senescent cells [154,155]. Cytosolic DNA molecules derived from retroelements elicit inflammatory responses [17]. In senescent cells, retrotransposable elements become transcriptionally activated. Thus, increased cytosolic LINE-1 cDNA may be a source of cGAS-STING activation in senescent cells [86]. Moreover, age-associated decline in mitotic fidelity gives rise to micronuclei formation, SASP expression, and cellular senescence [76]. Thus, cellular senescence reflects the axis of genome instability, misplaced cytosolic self-DNA, and inflammation, in the aging process (Figure 2).

## 5. Autophagy

Autophagy is an evolutionarily conserved process for the disposal of damaged organelles, protein aggregates, invading pathogens, and more [156]. In this process, cytosolic components are engulfed into membrane vesicles, the autophagosomes, which are subsequently delivered to lysosomes for degradation [156]. There are three types of autophagy defined by their types of cargo: macroautophagy, microautophagy, and chaperone-mediated autophagy. Here, we use autophagy for macroautophagy.

Studies, mostly done in animal models, show that autophagy activity declines with age. Moreover, experimentally generated defect in autophagy-related genes in mice, promotes aging phenotypes including disease susceptibility and immune failure [157]. Accordingly, in transgenic mice, an increase in basal autophagy extends lifespan and improves health span [158,159,160]. In humans, genetic studies have revealed the involvement of genes associated with autophagy in a variety of diseases including neurodegenerative, inflammatory, and autoimmune diseases [157].

There is a tight and highly conserved functional interaction between autophagy and the cGAS-STING pathway that defines the primordial function of STING and autophagy in cellular defense against pathogens [161,162]. Autophagy activity controls the accumulation of cytosolic self-DNA. An early study showed that autophagy compromised cells accumulated dysfunctional mitochondria which were prone to release of mtDNA following treatment with lipopolysaccharide (a component of gram-negative bacteria widely used to elicit immune response) [163], providing a mechanistic connection between autophagy failure and the susceptibility of mitochondria to release mtDNA. Accordingly, improved mitochondrial function by pharmacological stimulation of autophagy reduces age-associated inflammation [163,164].

Mutations in PTEN-induced putative kinase 1 (PINK1) and the cytosolic E3 ligase Parkin, cause familial Parkinson’s disease. Both PINK1 and Parkin are key factors in mediating selective mitochondrial autophagy called mitophagy. Intensive exercise induced a strong inflammatory phenotype in Parkin-KO and Pink1-KO mice that was rescued by ablation of STING [141].

It is evident that autophagy has key functions in mitochondrial quality control, immunity, and inflammation, and is essential for cellular homeostasis. The mode of the autophagy process, as well as the ramifications of autophagy failure, vary between different tissues and cell types, which may be further influenced by functional redundancy [157,165,166]. Such variations may determine the relative significance of misplaced cytosolic self-DNA in age-related inflammation in different tissues.

## 6. Sirtuins

Sirtuins are evolutionarily conserved protein deacetylases known to regulate aging and longevity across species [167]. Sirtuins use nicotinamide adenine dinucleotide (NAD^+^) as a co-factor to affect the function of a wide range of proteins [167]. There are seven sirtuins in mammalian cells (Sirt1-7). A family of enzymes that use NAD^+^ as co-substrate is poly (ADP-ribose) polymerases that bind to DNA damage and facilitate the recruitment of DNA damage repair proteins consuming NAD^+^ in the process [168]. The bioavailability of NAD^+^ is thought to decline with age [169,170], which may in part be a consequence of progressive increase in DNA damage in aging tissues [171,172].

Sirtuins control mitochondrial biology at several levels. SIRT3 maintains mitochondrial homeostasis by repair of non-enzymatic acetylation of mitochondrial proteins by acetyl-CoA [173]. More specifically, SIRT3 controls mitochondrial morphology by deacetylation of OPA1 [174], reduces oxidative stress levels by deacetylation activation of the mitochondrial superoxide dismutase, SOD2 [175], and contributes to mtDNA repair by direct interaction with the oxidative DNA damage repair OGG1 DNA glycosylase [176]. Peroxisome proliferator-activated receptor gamma coactivator-1-α (PGC-1α) has been described as a master regulator of mitochondrial biogenesis. SIRT1 controls mitochondrial biogenesis by PGC-1α deacetylation [177,178]. The SIRT1-PGC-1α signaling has recently been extensively reviewed [179].

The nuclear sirtuins SIRT6 and SIRT7 play key roles in DNA repair and genome stability. Accordingly, mice with experimentally ablated SIRT6 or SIRT7 genes suffer increased genome instability, accelerated aging phenotypes, and shortened lifespan [180,181,182]. SIRT6 and SIRT7 control retroelement LINE-1 activity by transcriptional repression [183,184,185]. Failure to repress LINE-1 activity in Sirt6-deficient mice resulted in the accumulation of LINE-1 cDNA in the cytoplasm followed by cGAS activation and IFNI production [183]. Suppression of expression of LINE-1 by SIRT6 seems to diminish with age as well as in senescent cells probably because of an increase in DNA damage response and declining cellular NAD^+^ levels [184]. In view of the key role of sirtuins in aging, mitochondrial homeostasis, and genome stability [167], further investigations may unravel important regulatory functions of sirtuins in controlling cytosolic-DNA driven inflammation in aging.

## 7. Segmental Progeroid Syndromes

Segmental progeroid syndromes, or premature aging syndromes is defined as genetic conditions that recapitulate some but not all aspects of physiological aging and display accelerated aging in one or more organs. Most cases of progeroid syndromes are caused by defect in specific genes involved in DNA homeostasis and DNA damage response [171].

Premature aging syndromes that are causally connected to defect in DNA repair and genome instability includes ataxia telangiectasia, Huntington-Gilford progeria [186], Bloom syndrome [187], Fanconi anemia [72], and xeroderma pigmentosum [188]. Inflammation is an integral part of the disease phenotype in all of these disorders, in some cases implicating various forms of misplaced cytosolic DNA and subsequent cGAS activation [77,189,190,191,192,193,194].

Mitochondria are reportedly implicated in the pathology of these diseases [195,196]. Interestingly, in ataxia telangiectasia, there is evidence for mtDNA release coupled to cGAS-STING activation [197,198].

Segmental progeroid syndromes represent “fast-forward” cellular genome instability, a hallmark of aging, and are as such appropriate models to investigate the connection between loss of genome instability and inflammation in the normal aging process. Notably, genome instability may develop over time, in part, because of altered and decline in the capacity of the cells to detect and to repair DNA lesions [57,170,199,200,201,202].

## 8. Cancer

Age is the major risk factor for increasing incidence of most cancer types. [203,204]. Accumulation of spontaneously occurring mutations in somatic cells that contribute to aging can also cause cancer [205]. Moreover, processes that drive the aging process and cellular transformation into cancer are largely controlled by shared mechanisms [203].

It has long been known that inflammation is an integral part of tumor progression and that various immune cells infiltrate malignant tumors [206,207]. Accordingly, cancer cells often develop mechanisms to avoid recognition and elimination by immune cells [208,209]. Chromosomal instability is an inherent characteristic of cancer cells and frequently results in the release of various forms of nuclear DNA, for instance, those originating from ruptured micronuclei, into cytoplasm [62,78,208].

The cGAS-STING signaling in cancer cells is an increasingly recognized and important mechanism connecting genome instability to immune cell infiltration and inflammation in neoplastic malignant tumors [210]. Subsequently, tumor cells frequently impair cGAS-STING signaling to evade immune surveillance [211,212,213]. Thus, agonists are being developed to restore and to enhance cGAS-STING activity and the subsequent antitumor immune response in cancer treatment [212]. It should be noted, however, that the immune response to cancer cells including the cGAS-STING signaling, is complex and an extensive review of the current state of the field is beyond the scope of this Review. The readers may consult other sources for in-depth reviews and studies [214,215,216].

## 9. Dynamic Modeling of Modulators of the cGAS-STING Pathway

Understanding the spatiotemporal characteristics of the cGAS-STING signaling and its connection to aging is essential for identifying effective therapeutic interventions. Computational modelling provides a complementary means to laboratory study for gaining this understanding.

At the molecular level, detailed structural information has been determined for key target molecules such as cGAS or STING. This information provides the essential input for generating molecular dynamics simulation models that can be used to study the interaction between targets and small molecule inhibitors. A recent study, successfully modelled the interaction of cGAS and several known inhibitors and offers, as noted by the authors, an efficient means to screen for new inhibitors from small molecule databases [217]. At the cell signaling network level, the dynamics can be simulated by using systems of ordinary differential equations (ODE) derived from the reactions within the network. Each reaction is described by the concentrations of the reactants and products together with the biochemical rate. ODE models are not ideal for spatial modelling but for essential compartmentalization, such as that between cytosol and nucleus. An ODE model with a network topology similar to that shown in Figure 1 has been developed and used to consolidate earlier findings that a severe knockdown in TREX1 is required to inhibit an inflammatory response to dsDNA [218]. ODE models have the advantage that they can be readily extended to incorporate new knowledge and data as it becomes available.

There are many such opportunities, which range from the relatively straightforward addition of post-translational modifications in key molecules to the more involved addition of newly discovered molecular mechanisms. For the cGAS-STING signaling, the recent finding that deactylation of cGAS controls full activation of the signaling network is an example of the former [219]. An example of the latter is the discovery that dsDNA and cGAS interaction may take place in liquid-like droplets [220]. Interestingly, this phase separation presents a barrier to the negative regulation of the cGAS-STING signaling by removal of the dsDNA by endonucleases such as TREX1 [221]. At the higher tissue level, simulation of processes such as inflammation requires a multicellular modelling approach. One approach is to embed ODE models within a population of single cells to enable simulation of multicellular systems. The ODE model mentioned above, has been adapted by the same group in a recent agent-based multicellular study [222]. Immune stimulatory DNA initiates signaling which drives an immune response via IFNI (IFNβ) expression and inflammation may spread by additional paracrine mechanisms (Figure 1). By varying initial concentrations to model cellular heterogeneity and production of IFNβ to model stochasticity, this model identified conditions limiting chronic inflammation, which are clearly important for aging. To the authors’ knowledge, no computational model has been reported that directly connects cGAS-STING signaling with aging, but the studies outlined above provide an important step in this direction.

Modelling at any level needs data for developing the model and importantly independent data to validate predictions made. The demands on data can be challenging but new technologies are constantly arising, one example being the development of BioSTING, a FRET-based biosensor of STING activity, for real-time observation [223]. Overall, a computational model that aligns well with data can be used to test interventions that would be infeasible with laboratory study alone. This is clearly helpful for drug development. Moreover, the spatiotemporal characteristics of cell signaling networks such as cGAS-STING and their impact on tissue and organisms over the life course is highly complex and computational modelling is a useful means to address this.

## 10. Concluding Remarks

The biology of chronic, age-related inflammation, is complex and is largely driven by stochastic damage to tissues and cellular components [7]. Thus, misplaced cytosolic self-DNA molecules originating from nuclear genome instability and alteration of systems that preserve the integrity of mitochondrial structure and function constitute a plausible source of age-related inflammation.

Self-DNA is frequently released into cytoplasm as evidenced, for instance, in TREX1 deficient patients and in Trex1^−/−^ mice showing constitutive innate immune activation and interferon production [17,51]. Archived formalin-fixed, paraffin-embedded tissue specimens contain important molecular information in their original state and location. A systematic immunohistochemistry analysis of cytosolic DNA and key proteins in the cGAS-STING pathway in post-mortem human tissues and in tissues from mice collected from different chronological age categories, may provide valuable in situ evidence for cGAS activation during physiological aging and warrants investigation.

Studies aiming at elucidating the cGAS-STING pathway as a potential source of chronic inflammation in aging should include analysis of the enzymes that prevent the accumulation of DNA molecules in the cytoplasm [17,224]. The catalytic activity, post-translational modifications, and spatiotemporal regulation, drive the proper function of these enzymes and may become compromised in aging tissues leading to a sluggish clearing of cytosolic self-DNA as, for instance, it seems to occur in senescent cells [225].

As we pointed out in Section 3.2.2, our current understanding of the role of cytosolic DNA in inflammation and the cGAS-STING pathway largely relies on cell lines and experimental settings in which the integrity of mitochondria is intentionally compromised. Obviously, more in vivo research is needed to document the physiologically relevance of these findings.

Delineating the underlying mechanisms for cytosolic self-DNA misplacement and the subsequent cellular immune responses, will increase our understanding of basic biology of the age-related inflammation. Moreover, such studies may provide useful mechanistic insights to develop therapeutic opportunities against inflammation in aging and in age-related disorders. Indeed, evidence is emerging that the cGAS-STING pathway is malleable to modification by pharmacological intervention [226].

## Figures and Tables

**Figure 1 cells-10-03544-f001:**
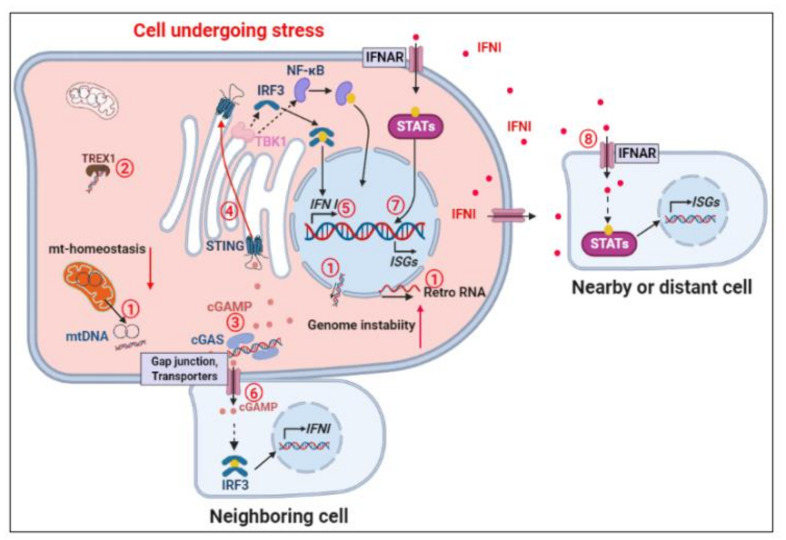
Activation and propagation of the cGAS-STING signaling by cytosolic self-DNA. Release of DNA from the nucleus and mitochondria as well as release of retroelement RNA molecules (Retro RNA) into cytoplasm in cells undergoing stress (1) can exceed the capacity of TREX 1 to clear cytosolic DNA (2). This will activate cGAS and the synthesis of the secondary messenger cyclic dinucleotide cGAMP (3) that binds to STING and results in its transport from the ER to the Golgi (4) subsequently leading to the production of Type I interferons (IFNI) (5). cGAMP can move into the neighboring cells through GAP junctions and specific cGAMP translocators and activate IFNI genes there (6). IFNIs subsequently activate the expression of interferon stimulated genes (ISGs) via paracrine and endocrine activation of specific IFNI membrane receptors (IFNAR) on the infected (7) as well as uninfected nearby and distant cells (8) that collectively promote the innate and adaptive defense responses. Red filled circles, IFNI. The Figure was created in BioRender.

**Figure 2 cells-10-03544-f002:**
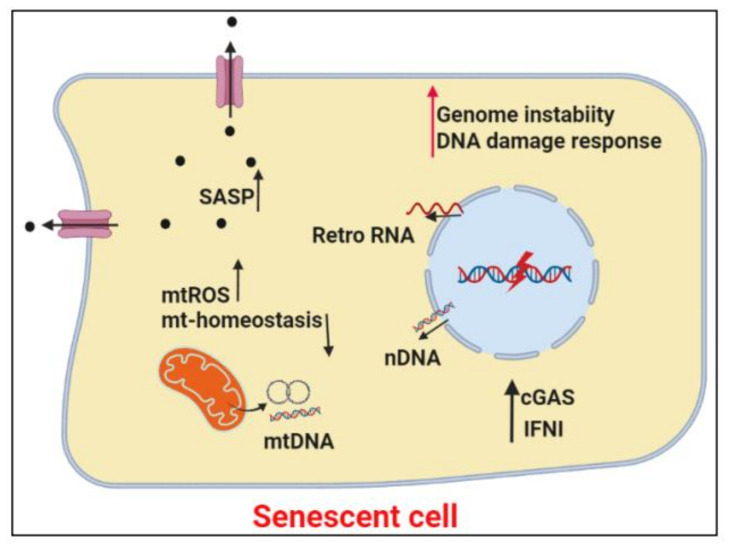
The intricate network of genome instability and inflammation in cellular senescence. Senescent cells typify the connection between genome instability and inflammation. cGAS-STING activation is key in the expression of senescence-associated secreted phenotype, SASP, that defines cellular senescence. Black filled circles, SASP; Retro RNA, Retroelement RNA. The Figure was created in BioRender.

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
