# Peer review of "Cytosolic Self-DNA—A Potential Source of Chronic Inflammation in Aging"

_cells, 2021, doi:10.3390/cells10123544_

Round 1

Reviewer 1 Report

Akbari and colleagues review a potential role of cytosolic self-DNA in chronic low-grade inflammation in ageing. The manuscript is overall very well written and a timely summary of the knowledge in the field.

Some minor points should be considered:

Line 43: “Inflammation can also be initiated in the anticipation of homeostasis disturbances like misplaced cellular components” is unclear and needs elaboration.

Line 157: “Retrotransposon DNA sequences are abundant in the human genome” is correct, but it seems that retrotransposition events are rare.

Paragraph Lines 239ff: ALS and frontotemporal dementia are not typically age-related diseases. Onset in familiar cases is around 50 years.

Line 294: This view has been questioned recently (https://doi.org/10.1016/j.cmet.2020.05.002)

Line 373: Sirt1-PGC-1 signaling has been reviewed recently (https://doi.org/10.3390/cells9081882)

Line 414: The link between immune cell infiltration and cancer progression is over-simplified e.g., one must consider MDSC and Treg function, NK activity, macrophage polarization, and endothelial cell adhesion/transmigration for the overall anticancer immune response (https://doi.org/10.1038/ncomms6852, https://doi.org/10.3390/cells9051133)

The concluding remarks should state explicitly the current limitations of our knowledge i.e., most of the data are from cell culture instead of in vivo models and the links to ageing in vivo are not so clearly defined.

The funding paragraph first sentence is incomplete.

Author Response

Comments from Reviewer 1:

Comment 1:

Line 43: “Inflammation can also be initiated in the anticipation of homeostasis disturbances like misplaced cellular components” is unclear and needs elaboration.

Response: Please see the new line: Inflammation can also be initiated by conditions that are often prelude to homeostasis disturbances like misplaced cellular components.

Comment 2:

Line 157: “Retrotransposon DNA sequences are abundant in the human genome” is correct, but it seems that retrotransposition events are rare.

Response: We have made this change: Activation of retrotransposons in mammalian cells is thought to be rare likely due to various control mechanisms including epigenetic suppression of their transcription.

Comment 3:

Paragraph Lines 239ff: ALS and frontotemporal dementia are not typically age-related diseases. Onset in familiar cases is around 50 years.

Response: New line: “Thus, although amyotrophic lateral sclerosis and frontotemporal dementia are not strictly age-related diseases, they demonstrate that cytosolic localization of mtDNA may account……”

Comment 4:

Line 294: This view has been questioned recently (https://doi.org/10.1016/j.cmet.2020.05.002).

Response: we have added: “Accordingly, selective clearance of senescent cells in mice has been reported to postpone aging-related disorders (149-151). However, indiscriminate elimination of senescent cells may disrupt tissue function and cause health deterioration because it appears that defined senescent cells may have important tissue specific functions (152).”

Comment 5:

Line 373: Sirt1-PGC-1 signaling has been reviewed recently (https://doi.org/10.3390/cells9081882).

Response: we have included “The SIRT1-PGC-1α signaling has recently been extensively reviewed (180).”

Comment 6:

Line 414: The link between immune cell infiltration and cancer progression is over-simplified e.g., one must consider MDSC and Treg function, NK activity, macrophage polarization, and endothelial cell adhesion/transmigration for the overall anticancer immune response (https://doi.org/10.1038/ncomms6852, https://doi.org/10.3390/cells9051133).

Response: We added these lines to that paragraph: It should be noted, however, that the immune response to cancer cells including the cGAS-STING signaling, is complex and an extensive review of the current state of the field is beyond the scope of this Review. The readers may consult other sources for in-depth reviews and studies (215-217).”

Comment 7:

The concluding remarks should state explicitly the current limitations of our knowledge i.e., most of the data are from cell culture instead of in vivo models and the links to ageing in vivo are not so clearly defined.

Response: We added the following paragraph to the Concluding remarks: “As we pointed out in section 3.2.3, our current understanding of the role of cytosolic DNA in inflammation and cGAS-STING pathway has largely relied on cell lines and experimental settings in which the integrity of mitochondria was intentionally compromised. Obviously, more in vivo research is needed to document the physiologically relevance of these findings.”

Comment 8:

The funding paragraph first sentence is incomplete.

Response: Corrected.

Reviewer 2 Report

Manuscript by Akbari M et al. concerns cytosolic self-DNA as a potential source of chronic inflammation in aging. Although, the subject area of the manuscript of some interest, it is intriguing that the authors did not discuss the role of cytosolic self-DNA levels-dependent differential activation of the cGAS-STING versus AIM2 inflammasome in chronic inflammation in aging. Further, the authors did not discuss whether activation of the inflammasome regulates the cGAS-STING pathway and production of type I IFN-β. It is known that both cGAS and STING proteins are type I IFN-inducible proteins. Therefore, Figure-1 needs to include a feedforward pathway to indicate the role of cytosolic self-DNA in up-regulation of both cGAS and STING protein levels and activation of the cGAS-STING pathway.

cGAS in certain settings requires the type I IFN-inducible IFI16 proteins for its sensing of cytosolic DNA and activation of the cGAS-STING pathway. Further, the IFI16 protein regulates the levels of STING. Notably, IFI16 protein has a role in sensing cytosolic self DNA, inflammation, and aging.

Minor:

Line: 45: “The age-associated chronic inflammation will damage tissues and organs and increase frailty and susceptibility to 45 disease (12,13).” Change to “The age-associated chronic inflammation damages tissues and organs and increases the frailty and susceptibility to disease (12,13).”

Line 145: “Micronuclei formation reportedly increases with age and appear to occur often in several accelerated aging syndromes (66-68).” Also state that cancer cells also produce micronuclei.

Line-185: “----networks and undergo----.” delete “and”.

Author Response

Comments from Reviewer 2:

Comment 1:

“…it is intriguing that the authors did not discuss the role of cytosolic self-DNA levels-dependent differential activation of the cGAS-STING versus AIM2 inflammasome in chronic inflammation in aging.”

Response: We are of course aware of AIM2. However, because AIM2 is largely linked to inflammasome activation and pyroptosis cell death, and the primary concept of this Review is based on low-grade chronic inflammation, and also because of space limitation, we confined to referring to published AIM2 and inflammasome related papers (references 19 and 149 in the original draft). We have now included the following lines to Section 2: “Several cytosolic DNA sensors have been identified including absent in melanoma 2 (AIM2), cyclic GMP-AMP synthase (cGAS), interferon-gamma inducible 16 (IFI16), and DNA-dependent activator of IRFs (DAI). Following binding to cytosolic DNA, these sensors activate various inflammatory responses (28).” And “AIM2 activation is best known to result in the assembly of the inflammasome (e.g. NLRP3), followed by activation of inflammatory caspases that trigger pyroptosis, a proinflammatory form of cell death (29). “

The “cGAS-STING pathway” is now in section 2.2.

Comment 2:

Further, the authors did not discuss whether activation of the inflammasome regulates the cGAS-STING pathway and production of type I IFN-β.

It is known that both cGAS and STING proteins are type I IFN-inducible proteins. Therefore, Figure-1 needs to include a feedforward pathway to indicate the role of cytosolic self-DNA in up-regulation of both cGAS and STING protein levels and activation of the cGAS-STING pathway.

Response: In our opinion, Figure 1 is already compact and adding more factors will only make it complicated. However, this is an important point and we have now added the following line to Section 2.2 at the end of second paragraph: “Moreover, cGAS-STING activation and response seems to be controlled in a feedback loop manner. Both cGAS and STING are reported as interferon stimulated genes (ISG) and their expression have been shown to be positively controlled by IFNI (47,48), thus enhancing the cGAS-STING response to cytosolic DNA.”

Comment 3:

cGAS in certain settings requires the type I IFN-inducible IFI16 proteins for its sensing of cytosolic DNA and activation of the cGAS-STING pathway. Further, the IFI16 protein regulates the levels of STING. Notably, IFI16 protein has a role in sensing cytosolic self DNA, inflammation, and aging.

Response: We have added the following lines to Section 2: “IFI16 senses cytosolic DNA from different sources and is able to activate IFNI as well as to trigger inflammasome assembly and pyroptosis (30-32). In human keratinocytes, IFI16 was found to functionally interact with cGAS for full activation of innate immune response to exogenous DNA and DNA viruses (33). Moreover, activated IFI16 promoted STING phosphorylation and transport (33). These results demonstrate that both the IFIT16 and cGAS DNA sensors trigger IFNI response through STING activation.”

Comment 3:

Line: 45: “The age-associated chronic inflammation will damage tissues and organs and increase frailty and susceptibility to 45 disease (12,13).” Change to “The age-associated chronic inflammation damages tissues and organs and increases the frailty and susceptibility to disease (12,13).”

Response: Done

Comment 4:

Line 145: “Micronuclei formation reportedly increases with age and appear to occur often in several accelerated aging syndromes (66-68).” Also state that cancer cells also produce micronuclei.

Response: Done. We added: “Micronuclei formation reportedly increases with age and appear to occur often in several accelerated aging syndromes (76-78), and in cancer (79).

Comment 5:

Line-185: “----networks and undergo----.” delete “and”.

Response. In our view, the conjunction “and” is necessary in this sentence.  

Reviewer 3 Report

Overall, this review is well written and gives an in depth mechanistic overview of the role cytosolic self DNA may play in inflammation and aging. This is a very interesting topic and very relevant to this important area of research. There are however some changes that should be made in order to enhance the translation aspect of this work. 

  1. In the abstract the authors mention the cytosolic self DNA- innate immune axis. However this is not discussed in the introduction. There should be a more clear hypothesis or schematic of this earlier on in the review.
  2. In the introduction it would be beneficial if certain inflammation related aging diseases that involve this pathway were discussed. 
  3. FIgure 1 should have numbers for each step on the figure which would help the reader follow this somewhat complex pathway.
  4. The organization of the paper should be somewhat reworked. I think if the authors. For example the authors start in 2. with the cGAS-STING pathway, but they have not yet done an explanation of cytosolic self-DNA. 
  5. All the sources cytosolic self-DNA are important and should be kept in the manuscript. However, the descriptions of each source are very detailed and the relationship of each of these sources to aging is listed at the end of each section with one sentence. One option is for the readers to have sections with different diseases and include each of these sources to the respective disease. Or a possibly more appropriate option is to add a paragraph to each source and better tying the mechanisms of each source to specific inflammation related diseases. 
  6. The section of mDNA mentions frailty, which is also mentioned in the beginning of the manuscript. However frailty itself and how it has inflammatory related components is never described. This is an important point which encompasses a large amount of aging related issues and should be further discussed. 
  7. The autophagy but never discuss other forms of cell death that are related to inflammation such as apoptosis or pyroptosis. It would be important to have a section on these. 
  8. There is a lack of specific innate immune pathways and how they are activated by cytosolic self DNA. This could be a section in itself or added to the appropriate sources. 
  9. On page 6 section 3, the authors discuss discuss neurons and mtDNA. This would be a very good place to introduce CNS related aging diseases such as AD. 
  10. Section 8 discusses cancer, however doesn't seem to get into the innate immune response in this section. Yet it is the only section that is disease-related. At the end of this section the authors mention modeling, but should include how this could be important for treatment.

In general, the manuscript seems to lose sight of the initial purpose which is to discuss the cytosolic self DNA- innate immune axis. If this is incorporated into more of the review then this will be a very interesting and important contribution to the field of inflammation and aging. 

Author Response

Comments from Reviewer 3:

Comment 1:

In the abstract the authors mention the cytosolic self DNA- innate immune axis. However this is not discussed in the introduction. There should be a more clear hypothesis or schematic of this earlier on in the review.

Response: We have reorganized the introduction as highlighted. Moreover, we have added a new paragraph to the introduction “Inflammation can also be initiated by conditions that are often prelude to homeostasis disturbances like misplaced cellular components (15). Thus, misplaced cytosolic nuclear and mitochondrial DNA, e.g.  as a result of impairment in the systems that maintain genome stability or those that are involved in mitochondrial quality control, have emerged as potent inflammation promoting cellular components by activating the innate immune system mediated by specific nucleic acid sensors as damage-associated molecular patterns (16).”

Comment 2:

In the introduction it would be beneficial if certain inflammation related aging diseases that involve this pathway were discussed.

Response: Please see the new introduction.

Comment 3:

FIgure 1 should have numbers for each step on the figure which would help the reader follow this somewhat complex pathway.

Response: Please see the revised Figure 1 and corresponding figure legend.

Comment 4:

The organization of the paper should be somewhat reworked. I think if the authors. For example the authors start in 2. with the cGAS-STING pathway, but they have not yet done an explanation of cytosolic self-DNA.

Response: Please see the revised introduction and the new Section 2.

Comment 5:

All the sources cytosolic self-DNA are important and should be kept in the manuscript. However, the descriptions of each source are very detailed and the relationship of each of these sources to aging is listed at the end of each section with one sentence. One option is for the readers to have sections with different diseases and include each of these sources to the respective disease. Or a possibly more appropriate option is to add a paragraph to each source and better tying the mechanisms of each source to specific inflammation related diseases.

Response: Our primary aim in this review is to introduce cytosolic DNA as a novel source of inflammation in aging tissues. In support of this model, we have briefly discussed some diseases.

Comment 6:

The section of mDNA mentions frailty, which is also mentioned in the beginning of the manuscript. However frailty itself and how it has inflammatory related components is never described. This is an important point which encompasses a large amount of aging related issues and should be further discussed.  

Response: We have added this paragraph to the Introduction: “Frailty commonly refers to a clinical state associated with increased vulnerability in older adults to risk of injury, disability, hospitalization, and mortality (8). Frailty is closely associated with inflammation and high levels of inflammatory marker including C-reactive protein and interleukin-6 have been reported in the pro-frail and frail elderly (9).”

Comment 7:

The autophagy but never discuss other forms of cell death that are related to inflammation such as apoptosis or pyroptosis. It would be important to have a section on these.

Response: Please see the revised section 2.

Comment 8:

There is a lack of specific innate immune pathways and how they are activated by cytosolic self DNA. This could be a section in itself or added to the appropriate sources. 

Response: Please see the new Sections 2 and 2.2

Comment 9:

On page 6 section 3, the authors discuss discuss neurons and mtDNA. This would be a very good place to introduce CNS related aging diseases such as AD.

Response: We have added the following paragraph: “Aging is the primary risk factor for most neurodegenerative diseases including Alzheimer disease (AD). Conditions that intimately associate with AD brains include amyloid β plaques, tau oligomerization and aggregation, mitochondrial dysfunction, and inflammation (121). In AD brains, intraneuronal accumulation of soluble tau oligomers correlates with synaptic alterations, neuroinflammation and memory deficits (122). Hippocampal neurons in a mouse model of human tauopathy, displayed cellular stress and altered mitochondrial morphology and homeostasis (120), as well as, T cell infiltration and inflammation that associated with cognitive decline (123).”

Comment 10:

Section 8 discusses cancer, however doesn't seem to get into the innate immune response in this section. Yet it is the only section that is disease-related.

Response: We appreciate these suggestions and comments, but an in-depth review of every section will exceed the permitted limits. As we have stated at the end of the Introduction “This Review is not an extensive description of these pathways. Sources of in-depth information are included for further reading.”

We have, however, added this lines at the end of Section 8: “ It should be noted, however, that the immune response to cancer cells including the cGAS-STING signaling, is complex and an extensive review of the current state of the field is beyond the scope of this Review. The readers may consult other sources for in-depth reviews and studies (215-217).”    

Comment 10:

At the end of this section the authors mention modeling, but should include how this could be important for treatment.

Response: This was a mistake. We somehow missed to number the “Dynamic modeling” part as a separate section. It is now section 9.

Round 2

Reviewer 2 Report

Authors have addressed my previous concerns. The manuscript appears much improved.